# Ocean precursors to the extreme Atlantic 2017 hurricane season

Samantha Hallam [1,2], Robert Marsh[2], Simon A. Josey [1], Pat Hyder[3], Ben Moat [1] & Joël J.-M. Hirschi[1]

Active Atlantic hurricane seasons are favoured by positive precursor sea surface temperature anomalies (SSTA) in the main development region (MDR, 10–20°N, 20–80°W). Here, we identify a different driving mechanism for these anomalies in 2017 (most costly season on record) compared to the recent active 2005 and 2010 seasons. In 2005 and 2010, a weakened Atlantic Meridional Overturning Circulation is the primary driver of positive SSTA. However, in 2017, reduced wind-driven cold water upwelling and weaker surface net heat loss in the north-eastern MDR were the main drivers. Our results are the first to show that air-sea heat flux and wind stress related processes are important in generating precursor positive SSTAs and that these processes were active pre-determinants of the 2017 season severity. In contrast to other strong seasons, positive SSTA developed later in 2017 (between April and July rather than March) compounding the challenge of predicting Atlantic hurricane season severity.

[1] National Oceanography Centre, European Way, Southampton SO14 3ZH, UK. [2] Ocean and Earth Science, University of Southampton, National Oceanography Centre, European Way, Southampton SO14 3ZH, UK. [3] Met Office, Fitzroy Road, Exeter EX1 3PB, UK. Correspondence and requests for materials should be addressed to S.H. (email: s.hallam@noc.soton.ac.uk)

Tropical cyclones (TCs) are one of the most costly natural hazards impacting on coastal regions across the world[1–3] and there is an increasing trend of TC related billion-dollar disasters in the US[4]. Over 70% of total tropical cyclone damage in the North Atlantic is caused by major TCs, category 3, 4 or 5 on the Saffir–Simpson scale, which make landfall[5].

In 2017, the North Atlantic hurricane season involved 6 landfalls and is the most costly season to date, with estimates of damage at over $360bn[6,7]. The season has broken several records. September had the largest Atlantic Accumulated Cyclone Energy (ACE) on record at 175, whilst the season ACE of 226 is the 7th highest. ACE (units, $10^4$ Kn$^2$) is defined as the sum of the maximum sustained surface wind speed squared at six-hourly intervals for all periods when the TC is at least of tropical storm strength ($>=34$ knots)[8]. The duration of hurricanes in 2017 was also record-breaking, setting a new September record of 41 hurricane days, principally due to Irma, Jose and Maria, each of which lasted for over 9 days.

The active season which unfolded was not, however, well predicted in the early season forecasts[9]. This suggests factors developed later in the season, leading to a higher level of activity than initially predicted. Here we compare the 2017 Atlantic hurricane season to the active seasons of 2005 and 2010 (as indicated in Fig. 1a). We explore the similarities and differences in the precursors to those seasons, predominantly from an ocean perspective, using a range of observational datasets. The seasons of 2005 and 2010 were chosen for their similarity in activity and intensity to 2017. The 2005 season has the 2nd highest Atlantic ACE on record at 250 with 15 hurricanes ($>=64$ knots) and 7 major hurricanes ($>=96$ knots). The 2010 season had an ACE of 165, 12 hurricanes (H) and 5 major hurricanes (MH) and was characterised by hurricanes with a significant duration, similar to 2017 (10 H, 6 MH), which formed further east. In addition, these seasons had the highest SSTAs in the MDR, between July and September for the period 1980–2017 (Fig. 1b).

As hurricanes intensify by extracting energy from the warm ocean surface, the underlying SSTs and ocean thermal structure are critical for their development[10] with local SSTs greater than 26.5 °C usually considered to be a necessary condition for tropical cyclone development, although it can vary slightly by ocean basin. Over 95% of TC in the Atlantic form in waters warmer than 25.7 °C[11–13]. Favourable (warm) ocean thermal structure in the MDR, together with a low vertical wind shear, and atmospheric low-pressure disturbances such as African Easterly Waves (AEW), are together conducive to intense and sustained hurricane development[5,9,11,14,15]. The size of the Atlantic Warm Pool (AWP), where water is warmer than 28.5 °C, has also been shown to influence the TC track, with more TC genesis further east for a large AWP. Its size also affects the position and strength of the Azores high[16]. When a large AWP is present, the Azores high weakens and shifts north eastwards, enabling TCs to track poleward and recurve towards the east.

Observed Atlantic hurricane frequency has been found to correlate with long-term variability of sea surface temperatures particularly as measured by the Atlantic Multi-decadal Variability (AMV) index[17–23], which is the North Atlantic area-averaged (0–60°N, 0–80°W) Sea Surface Temperature Anomaly (SSTA). In addition, positive Ocean Heat Content Anomalies (OHCA) have been found to increase the hurricane intensity and track length[24–26]. Variations in SST are influenced by the heat balance in the mixed layer of the ocean which is governed by air-sea fluxes, together with horizontal advection and vertical advection/mixing processes, and may be written as[13,27]:

$$\frac{\partial T}{\partial t} = \frac{Q_{net}}{(\rho C_p H)} + \left(\mathbf{U}_g + \mathbf{U}_{ek}\right) \cdot \nabla \mathbf{T} + \frac{(w_e + w_{ek})(T - T_b)}{H} \quad (1)$$

where $T$ is the mixed layer temperature (equivalent to the SST), $Q_{net}$ is the net surface heat flux i.e., the sum of the turbulent (sensible and latent) and radiative (solar and longwave) heat fluxes, $\rho$ is the density of seawater, $C_p$ is the specific heat of seawater, $H$ is the mixed layer depth, $\mathbf{U}_g$ is the geostrophic current velocity, $\mathbf{U}_{ek}$ is the Ekman current velocity, $w_e$ is the vertical entrainment rate, $w_{ek}$ is the Ekman pumping velocity and $T_b$ is the temperature of the water just below the mixed layer. The first term on the right hand side is determined by net surface heat flux, the second by horizontal advection and the third relates to vertical heat exchanges. Our analysis focuses on the respective contributions of surface heat fluxes and ocean circulation to the SST anomalies that can develop prior to hurricane seasons.

As an index of ocean heat transport, and convergence thereof, the observed Atlantic meridional overturning circulation (AMOC) strength at 26°N is used, as measured by the RAPID Array[28,29]. Previous studies[30–33] have identified links between ocean advection/AMOC strength and hurricane frequency, and also that the AMOC leads a SSTA dipole in the North Atlantic (with poles at 10–15°N and 45–60°N) that has maximum correlation strength at a 5-month lag[34]. Anomalies of the surface net heat flux (SFX) and wind stress curl (WSC) have also been analysed for their impact on SST although they have not previously been identified as precursors to hurricane season strength. A reduction in the WSC in the tropical Atlantic can lead to maximum positive SSTAs around 2 months later[35], while SFX anomalies directly warm or cool the surface layer[34,36–38].

Here we analyse SSTA alongside OHCA (using GODAS ocean re-analysis data[39]), and compare these to related indices and data. Our results reveal that the positive SSTAs prior to the 2017 active hurricane season were generated by processes not previously recognised to be important indicators of hurricane season strength. Furthermore, in contrast to other recent strong seasons, in which the SSTAs were evident in March, the anomalously warm ocean surface in 2017 developed later between April and July, making prediction of the Atlantic hurricane season severity even more difficult.

## Results

**Atlantic hurricane season activity.** The time series of Atlantic tropical cyclones from 1980 shows an increase in activity over the period (Fig. 1a). The number of landfalls has fluctuated with 6 or more recorded in 1985, 1996, 2005, 2008, 2010 and 2017. To better understand hurricane season activity, Fig. 1b highlights the ACE index for the years 1980–2017 and the associated SSTA and vertical wind shear anomaly in the MDR. Active seasons occur when there is a positive SSTA and a negative shear anomaly (i.e., weaker than average vertical wind shear) in the MDR (40% of years considered). Less active seasons occur when there is a negative SSTA and positive shear anomaly (43%) and when there is a positive SSTA and positive shear anomaly (14%). The seasonally averaged negative shear anomaly is greater in 2005 and 2010 compared to 2017. In both 2005 and 2010, negative anomalies persisted throughout the season, whereas in 2017 negative shear anomalies developed later in August and September. Previous work has linked ENSO variability / La Niña conditions with negative shear anomalies[40–43], SSTA[44–46] and a potential relationship between shear and SSTA[47]. We note that in 2005, 2010 and 2017 La Niña persisted during August and September. There is also a significant correlation between SSTA and shear anomaly of −0.58 ($p < 0.01$) indicating that positive (negative) SSTA are often associated with negative (positive) shear anomalies.

To composite the disparate influences on individual TC tracks (as opposed to season activity), Fig. 1c highlights conditions at 6

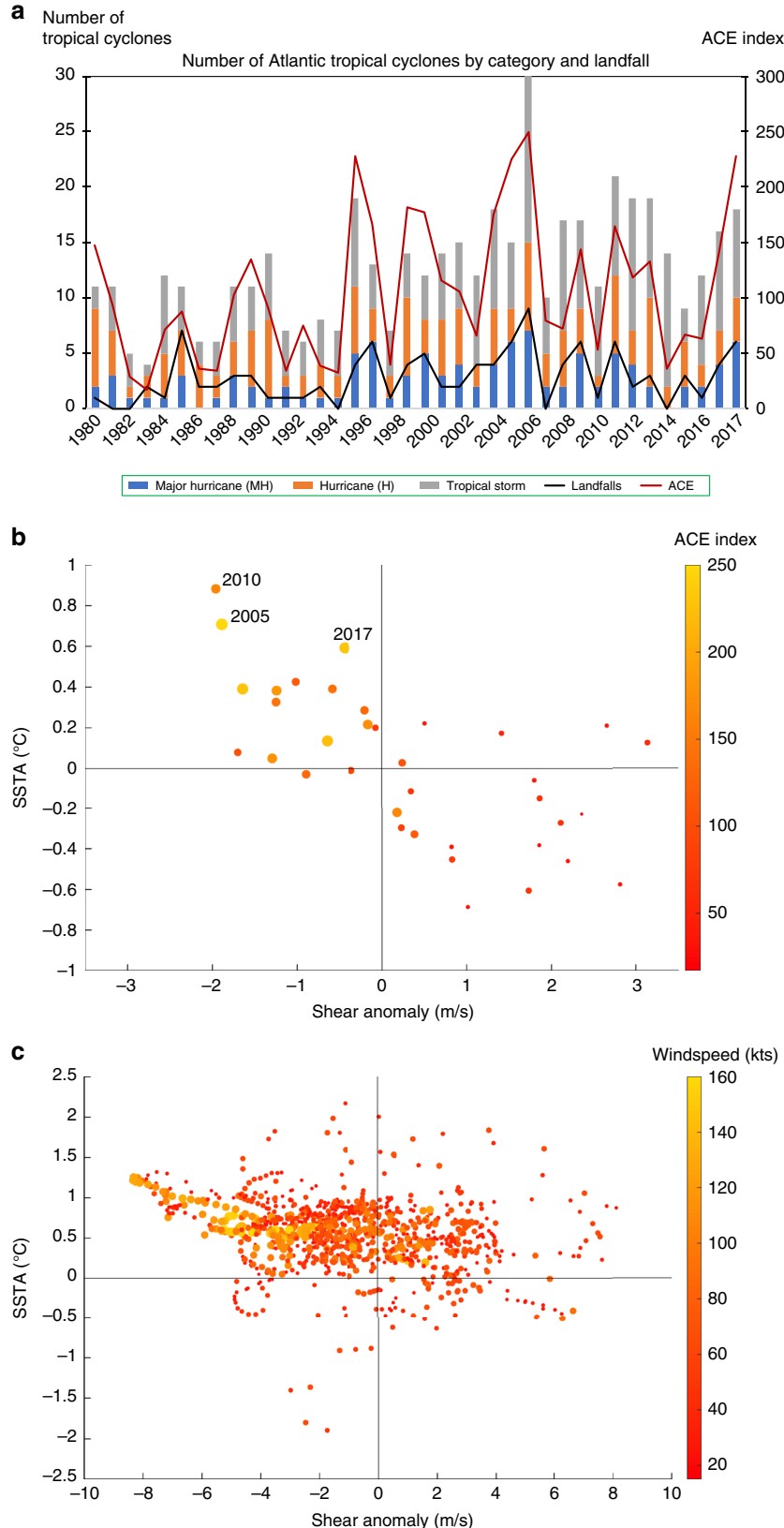

**Fig. 1** Atlantic tropical cyclone analysis. **a** Observed Atlantic tropical cyclone activity and ACE index from 1980–2017. **b** SSTA and vertical wind shear anomalies in the MDR between July and September for the period 1980–2017 and associated ACE Index (point size and colourbar). **c** Six hourly data from August to September Atlantic tropical cyclone tracks for the years 2005, 2010 and 2017, detailing the SSTA (°C), vertical shear anomaly, and associated wind speed (point size and colourbar)

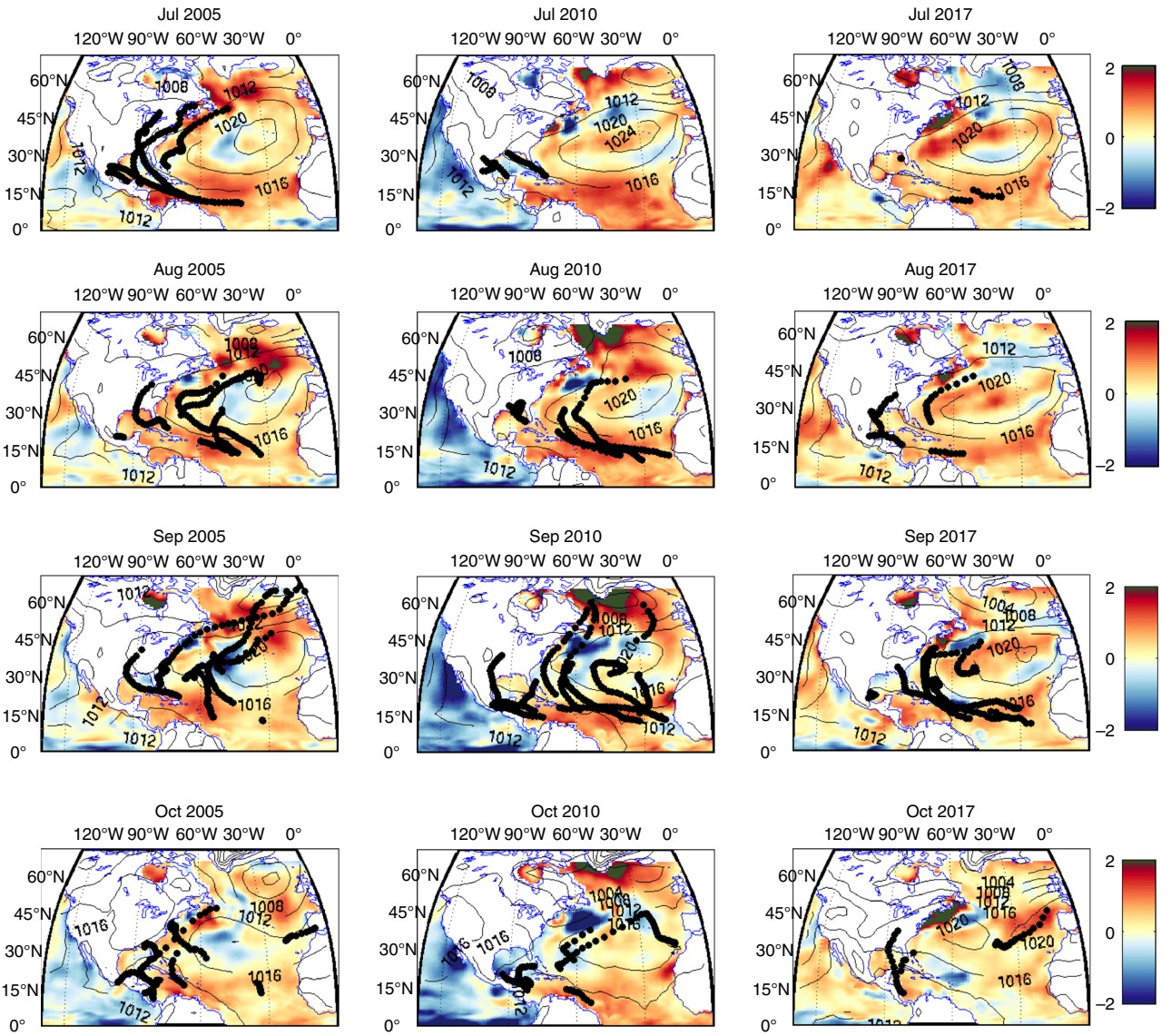

**Fig. 2** Atlantic tropical cyclone tracks during July, August, September and October for 2005, 2010 and 2017. Tropical cyclone tracks (black) and observed SSTA (colour, °C) overlaid with SLP (contours, mb)

hourly intervals along the TC tracks for August and September of 2005, 2010 and 2017, incorporating SSTA, vertical shear anomaly and wind speed along the track. Again, the most favourable conditions for TCs are positive SSTAs and a negative vertical shear anomaly as the majority of the 6 hourly values fall within the corresponding quadrant of Fig. 1c. A substantial number of values coincide with positive SSTAs and low positive vertical shear anomalies. Far less favourable for TC tracks are negative SSTAs and even when the wind shear anomalies are clearly negative we only find few hurricane track points in the corresponding quadrant. The relationships between SSTA and vertical shear, and number of tropical cyclones are significant, with correlations of 0.75 ($p < 0.01$) with SSTA and $-0.67$ ($p < 0.01$) with vertical shear between July and September, for the period 1980–2017. Figure 1c also highlights that wind speeds over 100 knots are only observed when there are positive SSTAs present and usually a negative vertical shear anomaly, indicating that both positive SSTA and weak vertical shear are critical to the full intensification of hurricanes. Previous studies have shown that 70% of major storms in the North Atlantic undergo rapid intensification[48] and the interplay between SSTA and vertical

shear in such intensification remains a matter for further research.

Of interest is the comparison between Fig. 1b and c, where the latter indicates TC tracks can exist where there are low positive values of vertical wind shear anomaly. The average vertical shear in the MDR in August and September is 6.14 ms$^{-1}$ (for the period 1948–2017). Wind shear magnitudes of less than 10 ms$^{-1}$ are generally considered favourable for TC genesis[9,14,49]; accordingly, positive shear anomaly values up to 4 ms$^{-1}$ are still likely to be conducive to TC development, consistent with a significant number of tracks observed in the top right quadrant (Fig. 1c).

Figure 2 shows SSTA overlaid with sea level pressure (SLP) and tropical cyclone tracks for 2005, 2010 and 2017. Positive SSTA in the MDR is evident in each year, although strongest in 2010. The 2005 season featured hurricane activity in all months, whereas in 2010 and 2017, September was the most active month. The 2010 and 2017 seasons were characterized by large SST anomalies, persisting across the MDR from 10–20°N and 15–90°W, from July to September. Accordingly, some tropical cyclone (TC) genesis was located further east than is usual in both years, and as far east as 20°W, in contrast to 2005. In 2005, the largest SSTAs

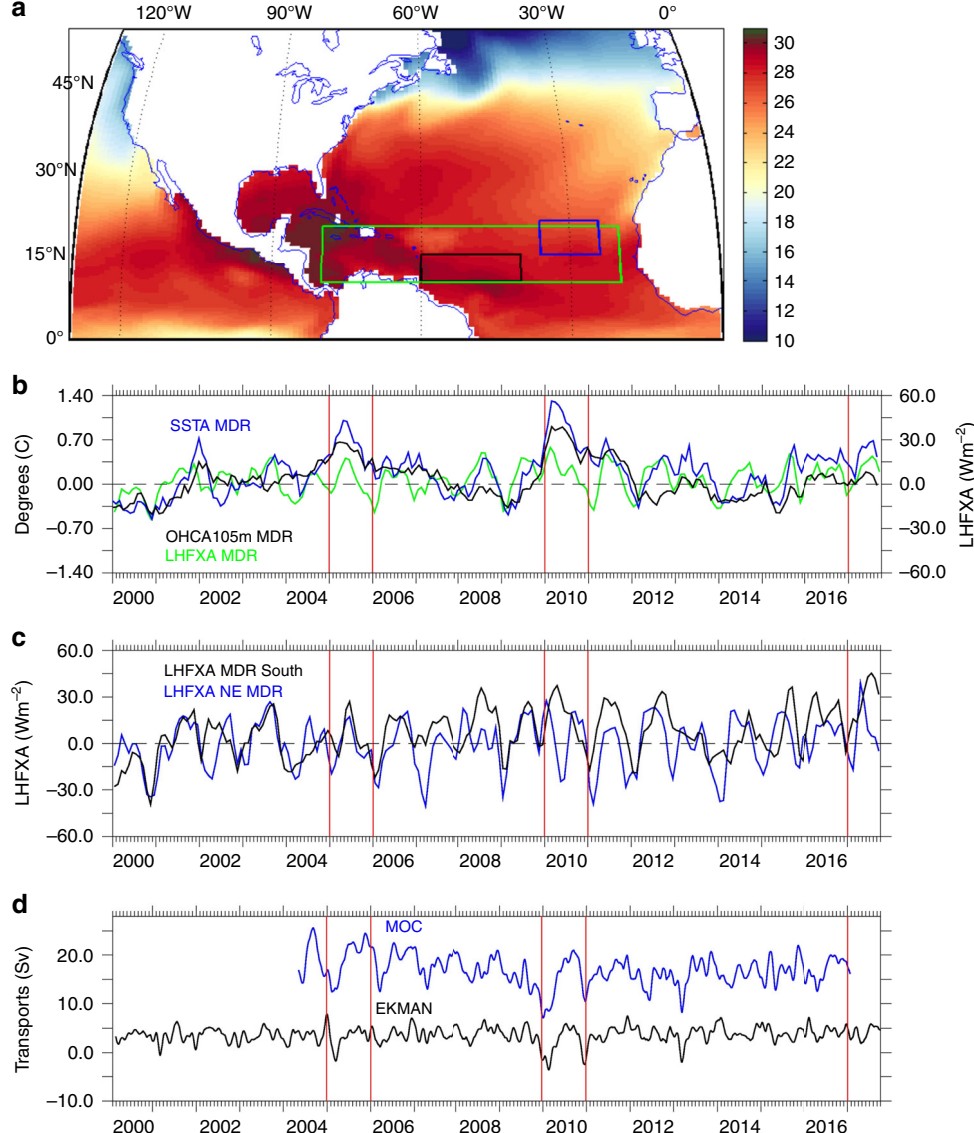

**Fig. 3** Atlantic time series and map outlining the study region. **a** SST September 2017: green box: MDR region 10–20°N and 20–80°W, black box: southern MDR region 10–15°N and 40–60°W, blue box: north-eastern (NE) MDR 15–21°N, 24–36°W. **b–d** Observed Atlantic time series: **b** monthly SSTA, OHCA105m and LHFXA in the MDR; **c** monthly latent heat flux anomalies (LHFXA) in the southern MDR and NE MDR smoothed over 3 months; and **d** MOC and Ekman Transports 12-hourly data smoothed over 61 days by applying a low pass filter

were further west at 40°W. In 2005 and 2010, positive temperature anomalies persisted to a depth of over 250 m during the season, whereas in 2017 the anomalies were generally only to a depth of 100 m between 10 and 60°W, although a positive anomaly to a depth of 250 m did exist during September 2017 between 60 and 80°W (Supplementary Fig. 1).

TC tracks are influenced by both SLP and SSTA. The positive SSTA across the MDR enabled an eastward shift in the genesis location in 2010 and 2017. In 2010, the Azores high was weaker and located further northeast in the Atlantic (especially in August), related to the large AWP[16], resulting in a lower pressure anomaly off the east coast of America (Supplementary Fig. 2) and enabling TC tracks to curve northwards. Accordingly, there were no US landfalls despite a very active season, and the 6 tracks which did make landfall in central America predominantly originated in the Gulf of Mexico or Caribbean Sea. In 2005 and September 2017, the lower pressure anomaly off the east coast of America during July to September also existed, but not to the

same extent as 2010. TC tracks did curve northwards but there were 9 and 6 landfalls in the respective years.

**Ocean and heat flux time series**. Turning to the ocean variables and to highlight the connection between SSTAs, air-sea fluxes and the ocean circulation we focus on the period from 2000 to 2017 (Fig. 3b–d). Ocean temperature variability is characterised by SSTAs and OHCA105 (temperature anomalies averaged over the top 105 m). The impact of air-sea fluxes on SSTA and OHCA105 is dominated by latent heat fluxes (LHFX). To understand the timing of the development of SSTAs in different parts of the MDR, LHFX anomalies (LHFXA) are either averaged over the entire MDR (Fig. 3b), the southern part of the MDR (10–15°N, 40–60°W) or the north-eastern (NE) MDR (15–21°N, 24–36°W) (Fig. 3c). The location of each region is shown in Fig. 3a. The north-eastern and southern regions of the MDR were chosen because the air-sea fluxes reached a maximum in these areas in

2017 during April and May–August, respectively. The Meridional Overturning Circulation (MOC) is the main contributor to ocean heat transport in the North Atlantic and here we use observations of the MOC and of its Ekman component at 26°N for the period from 2004 to 2017 (Fig. 3d).

SSTA and OHCA105m are closely aligned (Fig. 3b), although the strength of the anomaly is greater at the surface. Clear peaks in SSTA and OHCA105m are seen in 2005 and 2010, and also positive anomalies have been seen since 2015. LHFXA in the MDR (green line) is significantly correlated with SSTA and OHCA105m; correlation coefficients are respectively 0.52 and 0.38, both significant at the 0.01 level ($p < 0.01$). The LHFXA variability is analysed further in Fig. 3c for the two areas (southern MDR and NE MDR) that experienced the strongest anomalies in 2017. The most extreme latent heat flux anomaly values occurred in 2017, with latent heat loss weaker by 33 $Wm^{-2}$ sustained from May to August in the southern MDR region and weakening of over 60 $Wm^{-2}$ in the NE MDR during April (unsmoothed data). In 2005 and 2010, the LHFXA values in the southern MDR region were noticeably smaller in magnitude (May–Aug mean: 7 $Wm^{-2}$ in 2005; 13 $Wm^{-2}$ in 2010).

A more detailed analysis of the April and May–Aug net surface heat flux anomalies (SFXA) for the MDRs from 1980 to 2017 is shown in Supplementary Fig. 3. In the NE MDR, the 2017 April SFXA of 93 $Wm^{-2}$ was 3.4 standard deviations (SD) from the mean. In, the southern MDR, the 2017 May–August SFXA was 45 $Wm^{-2}$, 2.5 SD from the mean. For each MDR, the 2017 SFXA was the most extreme in the period considered. Furthermore, for the NE MDR the April 2017 SFXA was nearly twice that for the next largest anomaly (55 $Wm^{-2}$ in 2014) further emphasising the particularly unusual conditions prior to the 2017 season.

MOC values (Fig. 3d) are significantly below the seasonal mean values in February/March 2005 and 2010 at 11/13 Sv (1 Sv = $10^6$ $m^3 s^{-1}$) and 9/10 Sv, respectively. Average values for February and March are 15 Sv. Above average values are observed in February 2017 at 17 Sv (the latest available MOC data). Ekman transports are also below average in Feb/Mar 2005 and 2010 but are close to average in Feb/March 2017 and for the remainder of 2017. The February/ March values are important as Fig. 3b, d indicate that the observed MOC transport co-varies with the observed SSTA and OHCA105m in the MDR over the period from 2004 to 2017. The correlation between them is −0.35 for MOC-SSTA and −0.27 for MOC-OHCA105, when the MOC leads by 5 months, which is statistically significant at the 0.01 level ($p < 0.01$). This is in line with the results of Duchez et al.[34] who found the strongest correlation between the SSTA and MOC occurred when the MOC leads by 5 months. In addition, Supplementary Fig. 4 shows how the MOC transport anomaly in February and March co-varies statistically with SSTA and OHCA105m in July, August and September. Correlations over 0.5 are statistically significant at the 0.05 level ($p < 0.05$). Anti-correlations over 0.6 are seen over large areas of the MDR between 10–20°N and 30–70°W during July–September. Importantly, the correlations are higher for OHCA105m than SSTA, highlighting that the variability of the MOC transport at 26°N influences the upper ocean layer in the MDR, which is of consequence for hurricane intensification as it potentially provides a significant heat source extending over the top 100 m of the ocean to power hurricane development.

**Hurricane season precursors in 2005, 2010 and 2017.** Figure 4a shows the 2017 March–August monthly surface heat flux anomalies (SFXA) overlaid with the surface wind anomaly. In April, a strong positive SFXA of 100–150 $Wm^{-2}$ developed (i.e.,

300–350 $Wm^{-2}$ heat gain compared to the climatological mean of typically about 200 $Wm^{-2}$) with a maximum at over 150 $Wm^{-2}$ between 24–36°W and 16–18°N. This is associated with a reduction in the strength of the north easterly (NE) trade winds (30°W, 18–27°N) revealed by the wind vector anomalies. The weaker NE winds meant there was less cold, dry air over the region. As a consequence, the humidity gradient (between the sea surface and the overlying air) was lower than normal, and so the associated latent heat loss was reduced. In addition, the more humid air over the region enabled an increase in the downwards LongWave Radiative (LWR) flux. Anomalies in these two flux components largely account for the positive SFXA seen. Additionally, April 2017 is also characterised by a negative wind stress curl anomaly (WSCA) in the region associated with the anomalous winds (Fig. 4b). This results in a downward anomaly in Ekman pumping effectively reducing the upwelling of cold water at the eastern boundary and further assisting development of the positive SSTAs. Similarly, weaker than average NE/E winds persisted in June, July and August between 40–60°W, 9–15°N, along the southern boundary of the MDR. Again, the latent heat loss was lower and the LWR flux into the ocean was higher, largely explaining the positive SFXA and associated positive SSTA and OHCA in the southern MDR at this time.

The extent to which the 2017 SFXA generated the observed SSTA, and how this compares to 2005 and 2010, is also explored (Fig. 5). In 2005 and 2010, positive SSTAs already existed in the MDR in March (Fig. 5a, b). The patterns of these SSTAs are consistent with those related to the below-average MOC and Ekman transport in February and March generating the dipole SSTA pattern observed, with negative anomalies in the northern part of the Atlantic and positive anomalies in the south[32,34,50]. In 2017, in contrast, the MOC and Ekman transport was close to the 2004–16 mean, and SSTs were close to average (Fig. 5c). The SFXA forcing over April–July generates the SSTAs shown in Fig. 5d–f. In the MDR vicinity, SFXA-generated SSTAs are most substantial in 2017 (Fig. 5f), concentrated in the southern area of the MDR between 9–15°N and 30–60°W with a maximum around 1.4 °C.

Adding SFXA-generated temperature anomalies to the initial March SSTA patterns, we obtain estimates of the August SSTAs (Fig. 5g–i). When comparing the estimated to the observed SSTAs in August (Fig. 5j–l), the spatial SSTA patterns and sign in the MDR look similar, however the amplitude in 2005 and 2010 is higher than observed. Discrepancies (Fig. 5m−o) between surface flux based estimates and observations, can be attributed to the ocean circulation (advection) and mixing. The MOC recovered from its below-average strength in February and March during both 2005 and 2010 (Fig. 3d). This stronger MOC means that in spring and early summer more heat was transported northwards by the ocean, partly compensating for the MDR temperature changes linked to SFXA and assisting the development of average SST conditions in the north, whilst reducing the amplitude of the positive SSTAs in the south, seen during August in these years. Hence, the observed August SSTA in 2005 and 2010 are weaker than those estimated from surface heat flux alone. In 2017, the weaker NE/E trade winds observed from June to August in the MDR will have reduced the northward Ekman transport into the region. As less heat was transported north during those months it is likely to have led to the lower observed SSTA in August than predicted from surface fluxes alone. Additionally, along the eastern boundary between 9–18°N the weaker trade winds will have led to a reduced upwelling and the warmer observed SSTA in the area. From these results, we conclude that in 2005 and 2010, the initial ocean condition in March (MOC-related) was the main influence on SSTAs in August (Fig. 5j–l), whereas in 2017,

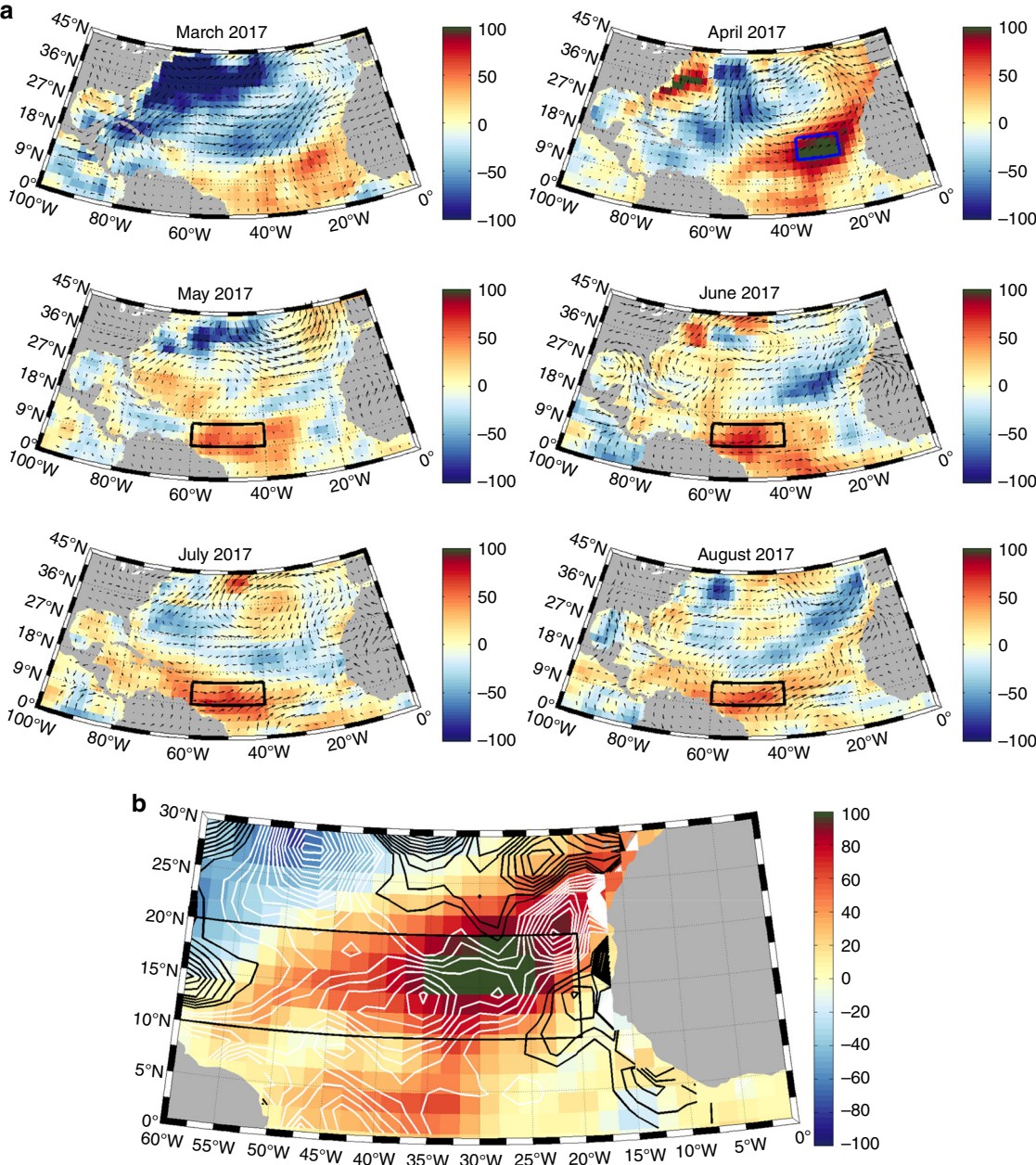

**Fig. 4** Atlantic Surface Heat Flux Anomalies (the sum of the net shortwave, net longwave, latent and sensible heat fluxes) during the build-up and early stages of the 2017 hurricane season. **a** Observed monthly surface heat flux anomaly from March to August 2017 (Wm$^{-2}$) overlaid with the 1000 mb wind anomaly, red colours indicate stronger heat gain than normal. Blue box indicates the eastern Atlantic region. Black box indicates the southern MDR region. **b** Observed Atlantic surface heat flux anomaly in April 2017 (Wm$^{-2}$) overlaid with the wind stress curl anomaly (WSCA). White contours indicate a negative WSCA (anomalous downwelling), grey contours indicate a positive or zero WSCA (anomalous upwelling). Contour units are 10$^{-8}$ Nm$^{-3}$

the SFXAs that developed from April to July were the dominant factor. These results are not specific to the GODAS ocean temperature reanalysis dataset used here and we found similar results with NCEP SST data (Supplementary Fig. 5).

Our analysis of a range of observations reveals the important role that surface heat fluxes played in positive SSTA development in the MDR in 2017, which was critical for subsequent hurricane activity in September. Figure 6 summarizes the mechanisms which were important during the season.

The reduction in the NE trade winds between April and July enabled positive latent heat flux and LWR anomalies to develop. The negative wind stress curl generated downward Ekman

pumping anomalies, suppressing the upwelling at the eastern boundary. These factors generated positive SSTAs. The reduction in the NE trade winds also reduced the Ekman transport of warm water into the MDR region, explaining why SFXA-predicted temperature anomalies are higher than observed, by August. Reduced vertical wind shear was also evident in late August and September, which together with the positive SSTAs, played a key role in the active hurricane season of 2017. In contrast, in 2005 and 2010, it was the reduced AMOC in February-March of those years which was the predominant cause of the positive SSTA in the MDR, which again combined with a weak vertical shear to favour very active hurricane seasons. These findings reveal for the

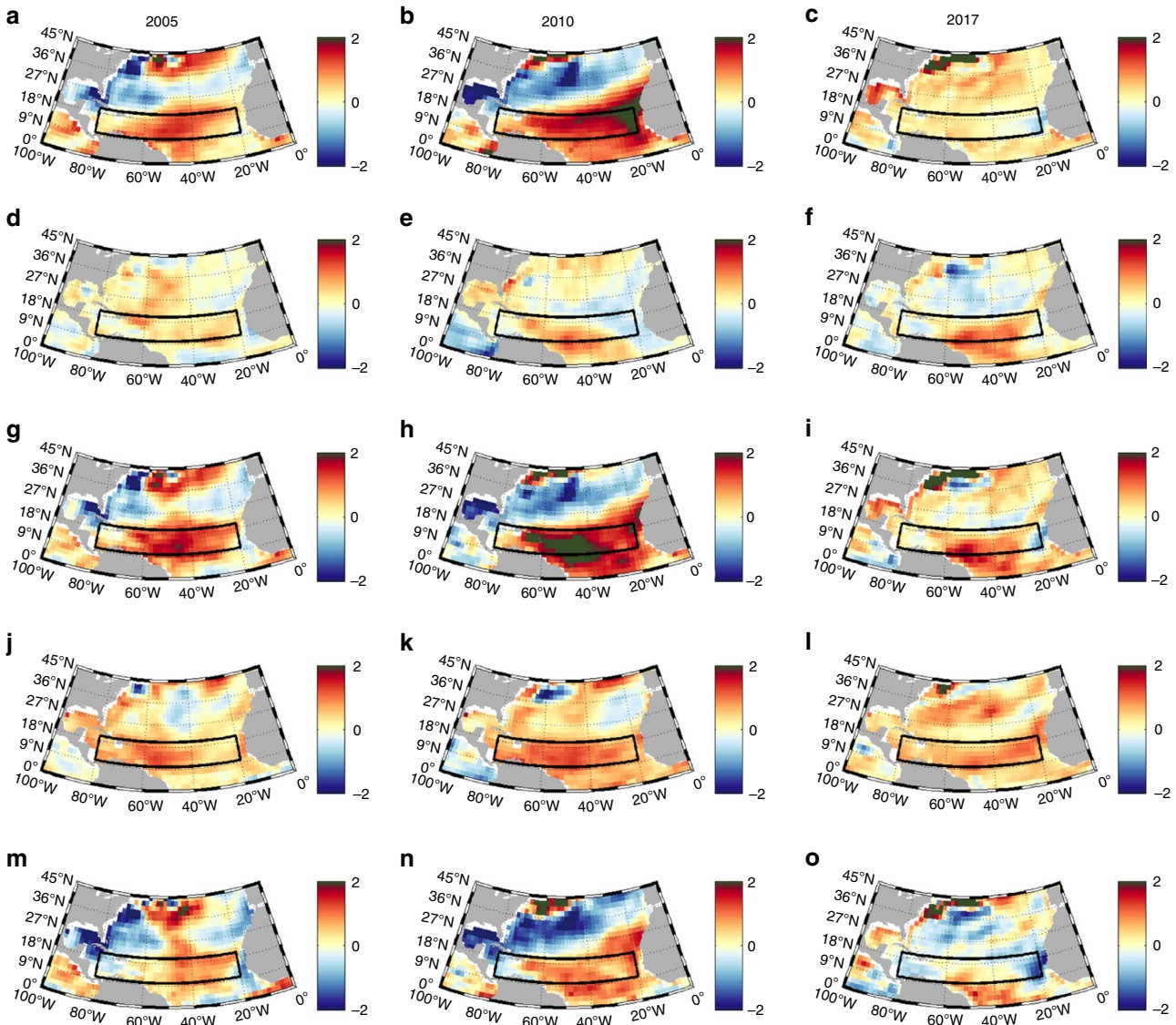

**Fig. 5** Surface Flux generated Temperature Anomaly April–July for 2005, 2010 and 2017. Initial condition—SSTA March (**a–c**). Estimated temperature anomaly April–July based on anomalous surface fluxes (**d–f**). Estimated SSTA in August formed by summing the initial condition and April–July surface flux generated temperature anomaly (**g–i**). Observed SSTA in August (**j–l**). Estimated minus observed August SSTA (**m–o**). Black box indicates MDR region. Colourbar units °C

first time that different precursors can generate positive SSTA in the MDR region, conducive to an active hurricane season. In 2017, surface fluxes were the dominant factor, whereas in 2005 and 2010 the AMOC played a key role.

In terms of hurricane prediction, the 2017 season was more difficult to forecast, as the surface flux anomalies developed between April and July shortly before the main season (August–September). For the other strong hurricane years considered, the reduction in the MOC / Ekman transport occurred earlier in February–March potentially enabling a longer lead time forecast given a sufficient observing system in place and noting also the importance of atmospheric conditions (vertical wind shear). In conclusion, our results have revealed that drivers of recent active hurricane seasons involving the ocean can take two forms: late winter changes in the ocean circulation and late spring/early summer changes in the air-sea heat flux. Developing forecast systems that adequately represent these processes will potentially aid preparedness and mitigation for the financial and societal consequences of hurricanes.

## Methods

**Hurricanes and ocean properties**. The observed Atlantic tropical cyclone and hurricane track data for the years 1980–2016 were obtained from HURDAT2, the revised Atlantic hurricane database[51]. Hurricane track data for 2017 was obtained from Unisys Weather (http://weather.unisys.com/hurricanes/). The NCEP Global Ocean Data Assimilation System[39] (GODAS) was used for the ocean temperatures at the surface and depth. All anomalies were based on the reference period 1980 to 2017 unless stated otherwise. The ocean heat content anomalies are based on the average temperature anomaly over the depth indicated. The observed AMOC strength and Ekman transport at 26°N, for the period 2004–2017, were obtained from the RAPID MOC monitoring project[52,53].

**Surface meteorology and air-sea exchanges**. The NCEP/NCAR reanalysis[54] was employed for sea level pressure (SLP), wind speeds and the air-sea heat flux. Wind speeds and associated anomalies were determined from the 1000 mb zonal and meridional components. The absolute vertical wind shear was calculated as the absolute difference between the 250-mb and 850-mb zonal wind using the reference period 1948 to 2017. The net surface heat flux (SFX) was determined as the sum of the net shortwave ($Q_{sw}$), net longwave ($Q_{lw}$), latent ($Q_{lat}$) and sensible ($Q_{sen}$) heat fluxes.

$$SFX = Q_{sw} + Q_{lw} + Q_{lat} + Q_{sen} \qquad (2)$$

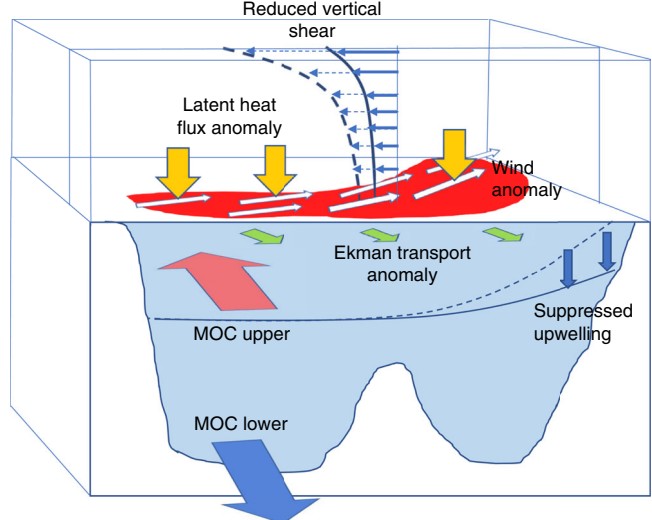

**Fig. 6** Schematic of the mechanisms which contributed to the positive SSTA in the MDR, and an active hurricane season, in 2017. Orange arrows are the heat exchange anomalies. White arrows show the wind driven anomalies forcing the ocean. The green arrows show the wind driven Ekman transport anomaly response. The blue line indicates the notional surface temperature with suppressed upwelling (solid line) and without suppressed upwelling (dashed line)

The estimated ocean temperature anomalies ($\Delta T$) used in Fig. 5b and c are based on the surface flux anomalies (SFXA) and calculated as follows:

$$\Delta T(x,y,t) = \Delta T(x,y,t_0) + \frac{1}{\rho D C_P} \int_{t_0}^{t} \text{SFXA}(x,y,t) \mathrm{d}t \qquad (3)$$

where $D$ = mixed layer depth (100 m), $\rho$ = density (1025 kg m$^{-3}$), $C_P$ = specific heat capacity 4182 JK$^{-1}$ kg$^{-1}$, $t_0$: April, $t$: July. A mixed layer depth of 100 m was chosen for consistency with the findings of Cayan[38], who showed the MLD in the North Atlantic to be between 150 m and 75 m between April and July. The spatial temperature pattern does not vary with depth chosen. A more detailed calculation would require a spatially and temporally varying MLD but the aim here is to show the magnitude of the heat flux related signal in line with the approach adopted by Duchez et al.[55] The wind stress curl was calculated from NCEP/NCAR reanalysis[54] monthly surface wind stress.

## Data availability

Hurdat2: https://www.nhc.noaa.gov/data/hurdat/; Hurricane track data 2017: http://weather.unisys.com/hurricanes/; GODAS: http://www.cpc.ncep.noaa.gov/products/GODAS/; AMOC and Ekman Transport data: http://www.rapid.ac.uk/rapidmoc; Sea Level Pressure and wind component data: https://www.esrl.noaa.gov/psd/data/gridded/data.ncep.reanalysis.html; Surface Flux data: https://www.esrl.noaa.gov/psd/data/gridded/data.ncep.reanalysis.surfaceflux.html.

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

## Acknowledgements

This work was supported by the Natural Environmental Research Council (NERC) [grant number NE/L002531/1], NERC projects DYNAMOC [NE/M005097/1] and ODYSEA (NE/M006107/1) and by the NERC programmes North Atlantic Climate System: Integrated Study (ACSIS) [NE/N018044/1], the RAPID-AMOC Climate Change (RAPID) programme, the European Union Horizon 2020 research and innovation programme BLUE-ACTION (Grant No. 727852). P.H. was supported by the Joint UK BEIS/Defra Met Office Hadley Centre Climate Programme (GA01101).

## Author contributions

S.H. led the development of the study, carried out the analysis and was lead writer of the paper. All authors contributed to the evolution of the analysis and to the writing of the paper.

## Additional information

**Competing interests:** The authors declare no competing interests.

