## [Peer Review File · Nature Communications]

Reviewer #1 (Remarks to the Author):

Overall, this is a well-written and well-organized manuscript that provides a novel look at the precursors that generated the SSTA in the MDR region that led to the active 2017 Atlantic Hurricane season. The authors compare the states that led to the active 2005 and 2010 seasons to that of the 2017 season and demonstrate that late spring/early summer changes in the air-sea heat flux were a key player in 2017 (which differs from 2005/2010). The authors present a well-thought-out mechanism for the 2017 series of processes that led to the active hurricane season. However given the difference from the 2005 and 2010 seasons, this leads me to ask the obvious question, is it possible that there are other active Atlantic seasons (perhaps with less landfalling storms, which may in part be random) during the recent record that display this late spring/early summer change in the air-sea heat flux? This would make the authors argument stronger. If not, can the authors provide more insight into how rare the 2017 precursors were? Once this main critique and the few minor comments below are taken into consideration, I believe that the manuscript is suitable for publication in Nature Communications after revisions.

Minor Comments:

L35-37: Is this for the North Atlantic region? Please specify.

L39-41: Again, the authors should mention that they are referring to the North Atlantic region.

L61: Year labels in figure 1b should be larger so that they are easier to read.

L135-137: I suggest the authors mention/reference some of the important literature on rapid intensification, which seems critical to this point. For example, Lee et al., 2016 (doi: 10.1038/ncomms10625) provides significant evidence that the majority of major hurricanes undergo rapid intensification.

L245: It is very difficult to see the WSCA contours in Figure 4b.

L311: Camp et al., 2018 suggested that they reasonably forecasted the spatial distribution anomaly of tropical cyclones with a May 15 seasonal forecast. Could you provide some insight, within your framework (i.e., did the model capture the SFXA?), as to why that might have been?

L336: Add the symbol for “delta T” after “ocean temperature anomalies”.

L484: Is Supplementary Figure 3 referred to in the main text?

Reviewer #2 (Remarks to the Author):

Summary

The manuscript by Hallam et al. presents an observational/reanalysis-based analysis comparing the atmospheric (primarily vertical wind shear) and oceanic factors that supported the active 2005, 2010, and 2017 Atlantic hurricane seasons. The main finding is that the 2017 season was unusual in that air-sea heat flux and wind-stress anomalies contributed to the SST warming favorable for tropical cyclones. This is an interesting and important topic, given the destructiveness of tropical cyclones and the relatively quick SST warming in 2017 that led to less skillful seasonal forecasts.

In my evaluation, the manuscript is publishable following major revisions. A better understanding of the oceanic factors that influence hurricane seasons is a valuable contribution, but I found that some parts of the analysis presented in the paper were unclear.

Major comments

- One of the major findings of this study is that air-sea heat flux and wind-stress processes can be important driving factors for the warm SSTs supportive of active hurricane seasons. What processes usually drive such SSTAs? What other processes could be important? Some equations that outline the relevant terms for upper-ocean heat content would be helpful.
- section on Ocean heat flux time series: It is unclear why the discussion switches back and forth between LHF_{SA} and MOC, and why the two are presented together in one (somewhat long) paragraph. Is there a connection between the two that is important to convey?
- It is also unclear why the sections on “ocean and heat flux time series” and “hurricane season precursors” are separate, as the former discusses LHF_X, which seems relevant to the SFXA discussion in the latter. It may be easier for a reader to understand if the sections are chosen according to ocean processes.
- The analysis on surface heat flux anomalies would also benefit from an equation that outlines the relevant terms. On line 243, the SFXA is attributed to specific components (which don't appear to be shown.). Similar comment for line 176, regarding the dominant contributions of LHF_X.
- The results on SST and wind shear influences on hurricanes present some well-known relationships, which is fine in establishing that both atmospheric and oceanic factors supported the three active hurricane seasons. If space is tight, this analysis/discussion could be made more concise, in favor of a more thorough (perhaps equation-based) analysis of the oceanic processes that generated the SSTAs, which seems to be the more interesting and novel part of this study.

Minor comments

- line 15: Suggest replacing “costliest ever” with “costliest on record”.
- lines 22-23: It is unclear how the two statements about “ positive surface net heat flux anomaly” are different.
- lines 39 and 49: “active and intensive” seems redundant.
- line 64: 26.5C is outdated due to overall warming.

- lines 98 and 203: The mention of trends isn't really supported by an analysis and seems not central to the paper's focus. Unless there is a strong reason to bring up trends, I suggest reconsidering whether to discuss them, and adding more substance if choosing to keep them in the discussion.

- line 186: What is the interpretation for the different LHF_{XA} in different regions?

- Figure 1: Suggest modifying the yellow-red colorbar, which didn't print well in my copy.

Reviewer #3 (Remarks to the Author):

Review of "Ocean precursors to the extreme Atlantic 2017 hurricane season" by Samantha Hallam, Robert Marsh, Simon A. Josey, Joel Hirschi, Pat Hyder and Ben Moat

This study investigated ocean precursors (i.e., MDR surface latent heat flux, MOC at 26.5N and MDR wind stress curl) to the active 2017 hurricane season. First, the study establishes the important roles played by the MDR SST and vertical wind shear anomalies on the active Atlantic hurricane seasons during 2005, 2010 and 2017. The highlight of this work is about the ocean precursors, namely surface latent heat flux and wind stress curl over MDR and MOC at 26.5N for the 2005, 2010 and 2017 seasons. The MDR surface latent heat flux anomalies are significantly correlated with MDR SST anomalies ($r \sim 0.52$). The MOC is also significantly correlated with the MDR SST anomalies ($r \sim -0.35$) when the MOC leads the MDR SST by 5 months. Consistent with this relationship, the MOC was weak in February-March of 2005 and 2010. The study also shows an overlapping of the positive surface heat flux and negative wind stress curl anomalies over the eastern tropical North Atlantic during April 2017, which suggests a possibility of reduced upwelling therein and thus contributes to the warm SST anomalies. The surface heat flux anomalies are further integrated in time to reconstruct mixed layer temperature anomalies in the North Atlantic in 2005, 2010 and 2017. A large (and mostly positive) difference between reconstruction and the observation indicates an active role of ocean dynamics and vertical mixing.

This is an excellent work. The study used only observational (and reanalysis) data to show convincingly that three ocean variables (i.e., MDR surface latent heat flux, MOC at 26.5N and MDR wind stress curl) are important precursors to active Atlantic hurricane seasons such as 2005, 2010 and 2017. Unfortunately, MOC data is available only until Feb 2017. So, it is difficult to apply this hypothesis for the 2017 hurricane season. I encourage the authors to contact Rapid Mocha team to see if the MOC data for the later period can be used for this paper. Another related suggestion is to update the Ekman transport at 26.5N shown in Figure 3d to the end of 2017, which may tell us about the post Feb/2017 MOC values. Other than that, I only have some minor editorial comments.

Minor comments:

1) Line 83: "...leads a SSTA dipole in the North Atlantic"

=>

It is not clear what “SSTA dipole in the North Atlantic” means. Describe it briefly here (e.g., where are the two poles located).

2) Lines 120-123: There is also a significant correlation between SSTA and shear anomaly of -0.58 ($p < 0.01$) indicating that positive (negative) SSTA are often associated with negative (positive) shear anomalies, which has been linked to ENSO variability.

=>

It is not very clear what has been linked to ENSO variability. Is it SSTA, vertical wind shear or the link between SSTA and wind shear? Please revise the sentence to make it clear. SSTAs in MDR have been linked to ENSO, NAO and Atlantic meridional mode (AMM) (e.g., Enfield and Major, 1997; Czaja et al., 2002; Lee et al., 2008). MDR SSTAs - vertical wind shear link was shown in Wang and Lee (2007). ENSO-vertical wind shear link was shown in Goldenberg and Shapiro (1996) and updated in Larson et al. (2012).

Goldenberg, S. B., and L. J. Shapiro (1996), Physical mechanisms for the association of El Niño and West African rainfall with Atlantic major hurricane activity, *J. Clim.*, 9, 1169–1187

Czaja, A., P. Van der Vaart, and J. Marshall (2002), A diagnostic study of the role of remote forcing in tropical Atlantic variability, *J. Clim.*, 15, 3280 – 3290

Wang, C. and S.-K. Lee, 2007: Atlantic warm pool, Caribbean low-level jet, and their potential impact on Atlantic hurricanes. *Geophys. Res. Lett.*, 34, L02703, <https://doi.org/10.1029/2006GL028579>.

Lee, S.-K., D. B. Enfield and C. Wang, 2008: Why do some El Ninos have no impact on tropical North Atlantic SST? *Geophys. Res. Lett.*, 35, L16705.

Larson, S., S.-K. Lee, C. Wang, E.-S. Chung and D. Enfield, 2012: Impacts of non-canonical El Niño patterns on Atlantic hurricane activity. *Geophys. Res. Lett.*, 39, L14706

3) Lines 178-179: “.... the southern part of the MDR (10-15°N, 179 40-60°W) or the north eastern MDR (15-21°N, 24-36°W)”

=>

Please explain here why these particular regions are selected to average surface latent heat fluxes. These two boxes are also used in the next section and Figure 4. It appears that these are the regions where surface flux was largest during 2017 hurricane season. Is that right? If so, can you use the same regions for other active hurricane seasons such as 2005?

4) Previous works on the ocean precursors:

The following paper discussed the importance of ocean advection on the 2005 MDR SST anomalies. This and others can be mentioned in the introduction or in discussion:

Foltz, G. R., & McPhaden, M. J. (2006). Unusually warm sea surface temperatures in the tropical North Atlantic during 2005. *Geophysical research letters*, 33(19).

Author's Response

"Ocean precursors to the extreme Atlantic 2017 hurricane season", Hallam et al.

We are grateful to all three reviewers for their thoughtful and insightful comments that have enabled us to significantly improve the manuscript. A detailed response on each point is provided below.

Reviewers' comments:

Reviewer #1 (Remarks to the Author):

Overall, this is a well-written and well-organized manuscript that provides a novel look at the precursors that generated the SSTA in the MDR region that led to the active 2017 Atlantic Hurricane season. The authors compare the states that led to the active 2005 and 2010 seasons to that of the 2017 season and demonstrate that late spring/early summer changes in the air-sea heat flux were a key player in 2017 (which differs from 2005/2010). The authors present a well-thought-out mechanism for the 2017 series of processes that led to the active hurricane season.

However given the difference from the 2005 and 2010 seasons, this leads me to ask the obvious question, is it possible that there are other active Atlantic seasons (perhaps with less landfalling storms, which may in part be random) during the recent record that display this late spring/early summer change in the air-sea heat flux? This would make the authors argument stronger. If not, can the authors provide more insight into how rare the 2017 precursors were? Once this main critique and the few minor comments below are taken into consideration, I believe that the manuscript is suitable for publication in Nature Communications after revisions.

The reviewer has raised an interesting point here. We have carried out analysis of the NCEP/NCAR reanalysis for 1980-2017 and find that the 2017 late spring/early summer heat flux anomalies are the most extreme positive values over the past nearly 40 years i.e. the weakest losses. We now provide more insight into how unusual these losses are through additional Supplementary Figure 3 and associated text in the main paper (section Ocean and heat flux time series).

Minor Comments:

L35-37: Is this for the North Atlantic region? Please specify.

Yes, over 70% of total tropical cyclone damage in the North Atlantic is caused by major TCs, category 3, 4 or 5 on the Saffir-Simpson scale, which make landfall. The text has been updated.

L39-41: Again, the authors should mention that they are referring to the North Atlantic region.

The text has been updated.

L61: Year labels in figure 1b should be larger so that they are easier to read.

The figure labels are now in a larger font.

L135-137: I suggest the authors mention/reference some of the important literature on rapid

intensification, which seems critical to this point. For example, Lee et al., 2016 (doi: 10.1038/ncomms10625) provides significant evidence that the majority of major hurricanes undergo rapid intensification.

The text now includes mention of rapid intensification of hurricanes and references have been included.

L245: It is very difficult to see the WSCA contours in Figure 4b.

The contour thickness has been increased for clarity.

L311: Camp et al., 2018 suggested that they reasonably forecasted the spatial distribution anomaly of tropical cyclones with a May 15 seasonal forecast. Could you provide some insight, within your framework (i.e., did the model capture the SFXA?), as to why that might have been?

The reviewer raises an interesting point but to answer in detail regarding capturing the SFXA would require analysis of the Camp et al., model which we are not in a position to do. However, we do now note that Camp et al., find better representation of El Nino and this may have been a further factor in their reasonable forecast of the cyclone spatial distribution.

L336: Add the symbol for “delta T” after “ocean temperature anomalies”.

The text has been updated.

L484: Is Supplementary Figure 3 referred to in the main text?

Supplementary Figure 3 is now Supplementary Figure 2 and is referred to in the text in line 164.

Reviewer #2 (Remarks to the Author):

Summary

The manuscript by Hallam et al. presents an observational/reanalysis-based analysis comparing the atmospheric (primarily vertical wind shear) and oceanic factors that supported the active 2005, 2010, and 2017 Atlantic hurricane seasons. The main finding is that the 2017 season was unusual in that air-sea heat flux and wind-stress anomalies contributed to the SST warming favorable for tropical cyclones. This is an interesting and important topic, given the destructiveness of tropical cyclones and the relatively quick SST warming in 2017 that led to less skillful seasonal forecasts.

In my evaluation, the manuscript is publishable following major revisions. A better understanding of the oceanic factors that influence hurricane seasons is a valuable contribution, but I found that some parts of the analysis presented in the paper were unclear.

Major comments

- One of the major findings of this study is that air-sea heat flux and wind-stress processes can be important driving factors for the warm SSTs supportive of active hurricane seasons. What processes usually drive such SSTAs? What other processes could be important? Some equations that outline the relevant terms for upper-ocean heat content would be helpful.

Variations in SST are influenced by the heat balance in the surface mixed layer, which comprises 3 main processes: surface fluxes, horizontal advection and vertical advection. The associated equation (1) is now included in the text with related discussion.

- section on Ocean heat flux time series: It is unclear why the discussion switches back and forth between LHFxA and MOC, and why the two are presented together in one (somewhat long) paragraph. Is there a connection between the two that is important to convey?

The addition of the equation for the heat balance in the surface mixed layer now makes it clearer why the two are presented together as they are the two key components influencing the upper ocean heat content and SSTA. The long paragraph has been split into two shorter ones for clarity.

- It is also unclear why the sections on “ocean and heat flux time series” and “hurricane season precursors” are separate, as the former discusses LHFx, which seems relevant to the SFXA discussion in the latter. It may be easier for a reader to understand if the sections are chosen according to ocean processes.

The goal of the section “ocean and heat flux timeseries” is to provide an overview of the relation between ocean heat content/SSTs and the MOC and LHFx for all years. To us this seems an intuitive way of introducing the section on “hurricane season precursors”, where we specifically zoom into the active seasons of 2005, 2010 and 2017 finding evidence of either ocean advection or LHFx being dominant. The section title has been updated accordingly.

We can see that the reviewer’s suggestion could be a reasonable alternative method of presentation but we prefer to retain our present structure as it enables the advection and surface flux contributions to be considered together.

- The analysis on surface heat flux anomalies would also benefit from an equation that outlines the relevant terms. On line 243, the SFXA is attributed to specific components (which don’t appear to be shown.). Similar comment for line 176, regarding the dominant contributions of LHFx.

The components of the net surface heat flux are now included within the description of the equation for the heat balance in the surface mixed layer and the relevant equation (2) is included in the methodology section.

- The results on SST and wind shear influences on hurricanes present some well-known relationships, which is fine in establishing that both atmospheric and oceanic factors supported the three active hurricane seasons. If space is tight, this analysis/discussion could be made more concise, in favor of a more thorough (perhaps equation-based) analysis of the oceanic processes that generated the SSTAs, which seems to be the more interesting and novel part of this study.

As noted above we have included a new equation (1) for the different processes involved to address the reviewer’s concern on this point. Since space is not tight we prefer to retain the analysis / discussion as it is rather than cutting it back.

Minor comments

- line 15: Suggest replacing “costliest ever” with “costliest on record”.

The manuscript has been updated.

- lines 22-23: It is unclear how the two statements about “positive surface net heat flux anomaly” are different.

The negative wind stress curl anomaly and positive surface net heat flux anomaly occurred in April in the north eastern part of the MDR, whereas the positive surface flux anomalies between May and August 2017 occurred in the southern MDR region. This is now clarified in the text.

- lines 39 and 49: “active and intensive” seems redundant.

The text has been updated.

- line 64: 26.5C is outdated due to overall warming.

The text has been updated to include more recent studies and reference to the North Atlantic.

- lines 98 and 203: The mention of trends isn’t really supported by an analysis and seems not central to the paper’s focus. Unless there is a strong reason to bring up trends, I suggest reconsidering whether to discuss them, and adding more substance if choosing to keep them in the discussion.

We understand the reviewer’s concern and the wording has been amended accordingly.

- line 186: What is the interpretation for the different LHF_{XA} in different regions?

The text has been updated to explain that the southern part of the MDR and north eastern MDR were chosen for further analysis because of the importance of these regions in 2017. Extreme LHF_{XA} were seen in April in the north eastern MDR, and between May and August in the southern MDR.

- Figure 1: Suggest modifying the yellow-red colorbar, which didn’t print well in my copy.

The colorbar has been modified to remove the lighter shades of yellow.

Reviewer #3 (Remarks to the Author):

Review of “Ocean precursors to the extreme Atlantic 2017 hurricane season” by Samantha Hallam, Robert Marsh, Simon A. Josey, Joel Hirschi, Pat Hyder and Ben Moat

This study investigated ocean precursors (i.e., MDR surface latent heat flux, MOC at 26.5N and MDR wind stress curl) to the active 2017 hurricane season. First, the study establishes the important roles played by the MDR SST and vertical wind shear anomalies on the active Atlantic hurricane seasons during 2005, 2010 and 2017. The highlight of this work is about the ocean precursors, namely surface latent heat flux and wind stress curl over MDR and MOC at 26.5N for the 2005, 2010 and 2017 seasons. The MDR surface latent heat flux anomalies are significantly correlated with MDR SST anomalies ($r \sim 0.52$). The MOC is also significantly correlated with the MDR SST anomalies ($r \sim -0.35$) when the MOC leads the MDR SST by 5 months. Consistent with this relationship, the MOC was weak in February-March of 2005 and 2010. The study also shows an overlapping of the positive surface heat flux and negative wind stress curl anomalies over the eastern tropical North Atlantic during April 2017,

which suggests a possibility of reduced upwelling therein and thus contributes to the warm SST anomalies. The surface heat flux anomalies are further integrated in time to reconstruct mixed layer temperature anomalies in the North Atlantic in 2005, 2010 and 2017. A large (and mostly positive) difference between reconstruction and the observation indicates an active role of ocean dynamics and vertical mixing.

This is an excellent work. The study used only observational (and reanalysis) data to show convincingly that three ocean variables (i.e., MDR surface latent heat flux, MOC at 26.5N and MDR wind stress curl) are important precursors to active Atlantic hurricane seasons such as 2005, 2010 and 2017. Unfortunately, MOC data is available only until Feb 2017. So, it is difficult to apply this hypothesis for the 2017 hurricane season. I encourage the authors to contact Rapid Mocha team to see if the MOC data for the later period can be used for this paper. Another related suggestion is to update the Ekman transport at 26.5N shown in Figure 3d to the end of 2017, which may tell us about the post Feb/2017 MOC values. Other than that, I only have some minor editorial comments.

The RAPID team have confirmed to us that MOC data for 26.5N for 2017 will unfortunately not be available until mid 2019 as the data still needs to be collected from the RAPID array at 26.5N and then analysed. However, Ekman transport data at 26.5N is now shown in Figure 3d to the end of 2017. There was no decrease in the Ekman transport during 2017, and in view of the close correlation between the MOC and Ekman transports as seen in Figure 3d, supports the conclusion that the MOC did not play a role in the development of the positive SSTA in the MDR region during 2017. This is now reflected in the manuscript.

Minor comments:

1) Line 83: "...leads a SSTA dipole in the North Atlantic"

=>

It is not clear what "SSTA dipole in the North Atlantic" means. Describe it briefly here (e.g., where are the two poles located).

The text has been updated to reflect the location of the dipole pattern with poles 10-15N and 45-60N.

2) Lines 120-123: There is also a significant correlation between SSTA and shear anomaly of -0.58 ($p < 0.01$) indicating that positive (negative) SSTA are often associated with negative (positive) shear anomalies, which has been linked to ENSO variability.

=>

It is not very clear what has been linked to ENSO variability. Is it SSTA, vertical wind shear or the link between SSTA and wind shear? Please revise the sentence to make it clear. SSTAs in MDR have been linked to ENSO, NAO and Atlantic meridional mode (AMM) (e.g., Enfield and Major, 1997; Czaja et al., 2002; Lee et al., 2008). MDR SSTAs - vertical wind shear link was shown in Wang and Lee (2007). ENSO-vertical wind shear link was shown in Goldenberg and Shapiro (1996) and updated in Larson et al. (2012).

Goldenberg, S. B., and L. J. Shapiro (1996), Physical mechanisms for the association of El Niño and West African rainfall with Atlantic major hurricane activity, J. Clim., 9, 1169–1187

Czaja, A., P. Van der Vaart, and J. Marshall (2002), A diagnostic study of the role of remote forcing in tropical Atlantic variability, J. Clim., 15, 3280 – 3290

Wang, C. and S.-K. Lee, 2007: Atlantic warm pool, Caribbean low-level jet, and their potential impact on Atlantic hurricanes. *Geophys. Res. Lett.*, 34, L02703, <https://doi.org/10.1029/2006GL028579>.

Lee, S.-K., D. B. Enfield and C. Wang, 2008: Why do some El Ninos have no impact on tropical North Atlantic SST? *Geophys. Res. Lett.*, 35, L16705.

Larson, S., S.-K. Lee, C. Wang, E.-S. Chung and D. Enfield, 2012: Impacts of non-canonical El Nino patterns on Atlantic hurricane activity. *Geophys. Res. Lett.*, 39, L14706

Thank you for the reference links which have all been included and were valuable to improving the manuscript as suggested.

3) Lines 178-179: "... the southern part of the MDR (10-15°N, 179 40-60°W) or the north eastern MDR (15-21°N, 24-36°W)"

=>

Please explain here why these particular regions are selected to average surface latent heat fluxes. These two boxes are also used in the next section and Figure 4. It appears that these are the regions where surface flux was largest during 2017 hurricane season. Is that right? If so, can you use the same regions for other active hurricane seasons such as 2005?

The text has been updated to explain that the southern part of the MDR and North eastern MDR were chosen for further analysis because of the importance of these regions in 2017. A supplementary Figure 3 has now been included which shows the net heat fluxes in these regions for each year from 1980, enabling a comparison with other active hurricanes seasons.

4) Previous works on the ocean precursors:

The following paper discussed the importance of ocean advection on the 2005 MDR SST anomalies. This and others can be mentioned in the introduction or in discussion:

Foltz, G. R., & McPhaden, M. J. (2006). Unusually warm sea surface temperatures in the tropical North Atlantic during 2005. *Geophysical research letters*, 33(19).

Thank you for the reference which has now been incorporated into the introduction.

Reviewer #1 (Remarks to the Author):

Again, the manuscript is well-written and well-organized. Below I have provided a few additional minor comments. Overall, the authors made significant improvements to the manuscript and I now believe it is ready for publication.

L184: If the MOC data used in Figure 3d is updated before final acceptance of the manuscript, it would be stronger if the analysis was shown through 2017.

L193: Is this really a weakening of 60 Wm^2 ? It doesn't appear that large in Fig. 3c. Or, perhaps the phrasing could make the explanation more clear!

L198: The caption of Supplementary Figure 3 needs to be updated so that it is clearer that the average regions (if that is indeed the case) are the same as those in Fig. 3c.

Reviewer #2 (Remarks to the Author):

The authors have addressed my comments very well and completely. I am happy to recommend this interesting and informative manuscript for publication. I just suggest the addition of two more references for links between ENSO and shear (line 123):

- Gray, W. M. (1984). Atlantic seasonal hurricane frequency. Part I: El Niño and 30 mb Quasi-Biennial oscillation influences. *Monthly Weather Review*, 112, 1649-1668.

- Patricola, C. M., Chang, P., & Saravanan, R. (2016). Degree of simulated suppression of Atlantic tropical cyclones modulated by flavour of El Niño. *Nature Geoscience*, 9, 155–160.